# Digital Health Platforms in Saudi Arabia: Determinants from the COVID-19 Pandemic Experience

**DOI:** 10.3390/healthcare9111517

**Published:** 2021-11-08

**Authors:** Saeed M. Alghamdi, Abdullah S. Alsulayyim, Jaber S. Alqahtani, Abdulelah M. Aldhahir

**Affiliations:** 1Respiratory Care Program, Clinical Technology Department, College of Applied Health Science, Umm Al Qura University, Makkah 21955, Saudi Arabia; 2National Heart and Lung Institute, Imperial College London, London SW7 2BX, UK; a.alsulayyim18@imperial.ac.uk; 3Respiratory Therapy Department, Faculty of Applied Medical Sciences, Jazan University, Jazan 45142, Saudi Arabia; aldhahir.abdulelah@hotmail.com; 4UCL Respiratory, University College London, London WC1E 6BT, UK; jaber.alqahtani.18@ucl.ac.uk; 5Department of Respiratory Care, Prince Sultan Military College of Health Sciences, Dammam 34313, Saudi Arabia

**Keywords:** digital health, health innovation, COVID-19

## Abstract

COVID-19 poses a significant burden to healthcare systems. Healthcare organisations with better health innovation infrastructures have faced a reduced burden and achieved success in curbing COVID-19. In Saudi Arabia, digital technologies have played a vital role in fighting SARS-CoV-2 transmission. In this paper, we aimed to summarise the experience of optimising digital health technologies in Saudi Arabia as well as to discuss capabilities and opportunities during and beyond the COVID-19 pandemic. A literature search was conducted up to September 2021 to document the experience of using DHTPs in Saudi Arabia in response to the COVID-19 outbreak. We also considered any published data, press briefings, and announcements by the MOH in Saudi Arabia. The findings were synthesised in narrative form. Health officials succeeded in optimising and maintaining a strategy to mitigate the spread of the virus via different digital technologies, such as mobile health applications, artificial intelligence, and machine learning. The quick digital response in Saudi Arabia was facilitated by governmental support and by considering users and technology determinants. Future research must concentrate on establishing and updating the guidelines for using DHTPs.

## 1. Introduction

Coronavirus (COVID-19) has already infected more than 186 nations and claimed thousands of lives, incapacitating even more. Responses to the pandemic have varied dramatically, and the demand on healthcare professionals has increased due to the disruption caused by the COVID-19 outbreak [1,2,3,4].

Around the world, healthcare officials are constantly reviewing appropriate public measures to prevent the spread of COVID-19. For example, minimising physical contact, quarantines, social distancing, and wearing masks have been implemented as measures to keep people safe from infection, especially vulnerable people with chronic conditions. Ref. [5] These public health measures have increasingly encouraged the use of technology to support daily activities and healthcare delivery [5].

In Saudi Arabia, the Ministry of Health (MOH) recently established a strategy to mitigate COVID-19. The MOH strategy includes using digital health technology platforms (DHTPs) to provide long-distance follow-up and control COVID-19 outbreaks [6]. Different terms express the application of DHTPs in healthcare delivery, and these are outlined in Table 1. The aim of this paper is to document the experience of using digital health technologies in Saudi Arabia and to discuss determinants for optimising DHTPs during the COVID-19 pandemic.

## 2. Methods

An electronic search of the following engines was conducted up to September 2021 to retrieve articles: MEDLINE (Ovid), CINAHL, and PubMed. The keywords and subject headings used in electronic search were: mobile application, telemedicine, telehealth, health technology, digital health platforms, COVID-19, and Saudi Arabia. The electronic search strategy and the results from the database are provided in Appendix A. We also considered any published data, press briefings, and announcements by the MOH in Saudi Arabia. The search included both sources in English and Arabic. The inclusion criteria included studies that critiqued the experience of using DHTPs in Saudi Arabia in response to the COVID-19 outbreak in Appendix A. Any article that evaluated DHTPs usability and/or not specific to Saudi Arabia was excluded. The findings were documented in narrative form. 

## 3. Results

Figure 1 shows the current DHTPs used during the COVID-19 outbreak in Saudi Arabia. In general, the strategy was to implement e-health and optimise DHTPs to control the spread of COVID-19 in Saudi Arabia. We documented the findings into three subheadings as follows: 

### 3.1. Digital Response in Saudi Arabia 

Globally, there has been a strong drive to implement and accelerate DHTPs, led by governments and health authorities to mitigate the spread of COVID-19. Evidence continues to emerge that using digital health strategies to provide and manage health services is a significant factor in stopping or controlling SARS-CoV-2 transmission. The use of DHTPs has also demonstrated a positive impact on reducing the pressure on healthcare systems and facilitated controlling COVID-19 [13,14].

In Saudi Arabia, the MOH has succeeded in optimising and maintaining a strategy to mitigate the spread of the virus via different digital health technologies [15]. This strategy includes screening suspected patients, providing daily reports about symptoms, tracing confirmed COVID-19 patients, and maintaining social distancing via advanced technology, such as mobile health applications, artificial intelligence, and machine learning (Figure 1). 

The development of DHTPs to improve the accessibility of healthcare services across the kingdom was already in place following the Saudi Vision 2030 plan, which stressed the importance of adopting and developing a national telehealth network [16,17,18].

Optimising DHTPs requires previous experience in the cycle of knowledge translations of health innovations and digital health technologies [17,19]. This experience has been reported in Saudi Arabia through previous studies, which described the potential and acceptability of DHTPs among the Saudi population. The evidence showed positive results, considering the challenges and limited perspective of clinicians, patients, policymakers, and stakeholders [3,20,21,22,23,24,25]. Previous national surveys in Saudi Arabia have demonstrated that the DHTPs for COVID-19 have the advantage of wide access, user-friendly interface, ease of use, and high acceptance rate [22,23,24,25,26,27].

### 3.2. Determinants for Optimising DHTPs

In response to the COVID-19 crisis, a multidisciplinary Saudi team from different sectors, including commercial partners, joined forces to overcome these challenges and make the DHTPs easily accessible to citizens and non-citizens in the kingdom of Saudi Arabia. 

Strong financial and logistic support from the MOH facilitated DHTPs within a short period of time. According to the MOH digital transformation report (covering the period 2019–2021), there were more than 23 million registered users on DHTPs such as Mawid, Sehha, Sehhaty, Tawakkalna, Tetamman, and Tabaud [28]. During the pandemic, it was obvious that many outpatient clinics in primary and tertiary hospitals in Saudi Arabia incorporated DHTPs, which facilitated monitoring, managing, and delivering non-urgent medical care in an practical way, thereby, maintaining public measures, such as self-isolating and social distancing [29] (Figure 2).

The MOH in Saudi Arabia developed and introduced additional features for current DHTPs to cope with the COVID-19 pandemic. These were effectively incorporated into the healthcare system. The DHTPs provided benefits as well as trained human support via free direct line 937, both technical and healthcare providers, was also available 24/7 and people were able to access services from any location in Saudi Arabia [3,30]. The strategy was not to replace the human factor with technology; it was to support the technology with a greater human factor to optimise its effectiveness in fighting COVID-19. 

Data availability was an essential determinant for optimising the DHTPs during the COVID-19 pandemic. Since patient zero was discovered, non-confidential data of COVID-19 have been available and free to use to prompt health innovations and digital health technology applications. Many researchers have used the data to model the future of the pandemic or to predict new cases and deaths in the upcoming months [31,32,33,34,35]. Data accessibility has facilitated building models to predict the future of the pandemic in Saudi Arabia and has helped decision-makers to set plans and strategies to fight COVID-19 [36,37].

The technical determinant of optimising DHTPs in Saudi Arabia was the robust infrastructure of both networks and technologies [38]. The MOH in Saudi Arabia effectively encouraged people to use DHTPs to receive care rather than visiting primary care clinics during the COVID-19 pandemic [3,17]. DHTPs were central to the role of primary health services to help with triage and mitigating the spread of COVID-19 [1,2,39,40].

Internet connectivity is widely available to the population in Saudi Arabia, which has supported the healthcare system to continually provide high-quality care for all patients in the private and public sectors [29]. Saudi Arabia is identifed as one of the countries in the Middle East where more than 90% of its population have Internet connections [41]. Big telecom companies in Saudi Arabia, such as Saudi Telecom Company (STC), Zein, and Mobily, announced free data use for the DHTPs during the pandemic, which ultimately facilitated high usage and smooth usability of the technologies [3].

### 3.3. DHTPs and Hajj Season 

A success story must be told about the Hajj season in 2021. With the help of DHPTs, the authorities in Saudi Arabia transformed the Hajj season from a super spreading event and COVID-19 variant farm into a safe and healthy season with zero COVID-19 cases [15,42].

Prior to the COVID-19 pandemic, health authorities in Saudi Arabia were prepared for any potential spread of infectious diseases associated with mass gatherings (e.g., Hajj season) [43]. The coinciding of COVID-19 with the Hajj season is an extra indication of the insights and benefits that DHTPs can provide, such as predicting the progression and the effects of COVID-19 during the Hajj season via mathematical modelling and machine learning. A recent study examined agent-based modelling for curbing COVID-19 spread during the Hajj season. The findings showed that control measures, such as buffers and face masks, were simple but had a significant impact on controlling infection, especially during the Hajj season. This mathematical modelling provides robust information to support decision-making about public measures [15,44].

One of the advanced health innovations launched during the Hajj was the “Pilgrim eBracelet” to monitor Hajj performers, screen their daily symptoms, and monitor mass gathering. The smart bracelet is connected to the Internet and healthcare professionals to provide several services, such as monitoring the Hajj performers’ health condition and any change in exposure to COVID-19 cases. The smart bracelet also provides information about heart rate and oxygen saturation. Moreover, the bracelet allows the Hajj performers to seek assistance in the case of an emergency. However, information about acceptance, performance, feasibility, and functionality of the “Pilgrim eBracelet” is limited and only news published that information. Last season, 5000 Pilgrim eBracelets were distributed to pilgrims as a trial version. Outcomes have not been published so far, but the season ended with zero infection cases. This link to DHTPs may have helped to design a safe and secure environment to perform Hajj without any unexpected cases of infection [15,45].

## 4. Conclusions

Health officials in Saudi Arabia have succeeded in optimising and maintaining a strategy to mitigate the spread of the COVID-19 virus via different digital technologies, such as mobile health applications, artificial intelligence, and machine learning. The quick digital response in Saudi Arabia has been facilitated by governmental support and considering users and technology determinants. Considering all of this success, we still need further work to sustain the use of DHTPs and explore the experience of artificial intelligence in guiding clinical practice. We also need to increase the interconnected network between primary and tertiary health centres to share electronic medical records with other healthcare centres for a better interpretation and management strategy.

## Figures and Tables

**Figure 1 healthcare-09-01517-f001:**
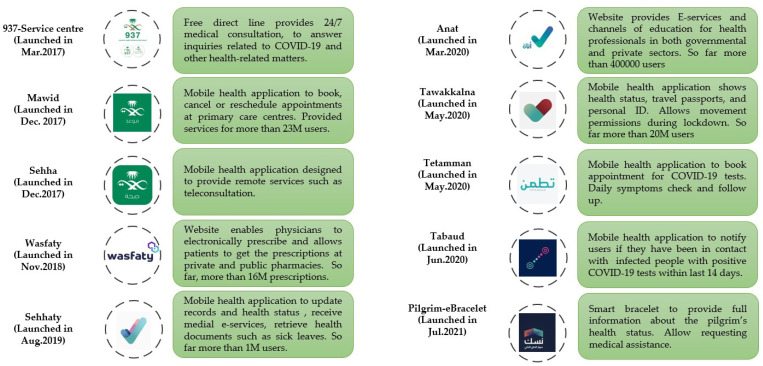
The timeline of optimising DHTPs in Saudi Arabia. Different DHTP applications were launched before, during, and after the COVID-19 pandemic.

**Figure 2 healthcare-09-01517-f002:**
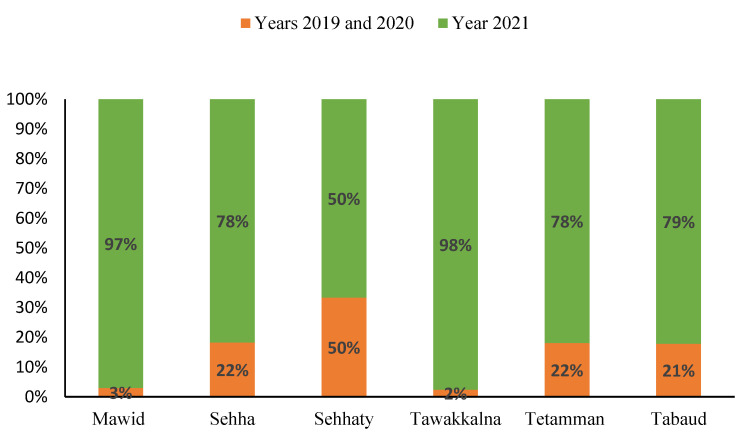
Increase in the number of registered users of DHTPs over the period from 2019 to 2021. Data presented as proportions [17,28].

**Table 1 healthcare-09-01517-t001:** The application of digital health technology platforms (DHTPs) during the COVID-19 pandemic.

DHTP Terminology	Application during COVID-19 Outbreak
Digital Health Technology Platforms (DHTPs)	Using information and communication technologies to provide distance health services for people as well as support decision-making processes for healthcare professionals [7].
Artificial Intelligence (AI)	Using machines to screen people in large public places, provide fast diagnosis, and detect infected people with fever [8,9].
Machine Learning (ML)	Using mathematical algorithms to predict infected cases, mortality, and vaccinations [10,11].
Mobile health applications	Using mobile apps to track daily symptoms of infected and non-infected people as well as book appointments and vaccinations [12].

## Data Availability

Not applicable.

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
