# Peer review of "Digital Health Platforms in Saudi Arabia: Determinants from the COVID-19 Pandemic Experience"

_healthcare, 2021, doi:10.3390/healthcare9111517_

Round 1

Reviewer 1 Report

The authors have acknowledged suggested revisions and made changes where appropriate and also justified when they did not find it appropriate to do so. However, a justification that was given by the authors was to keep the work specific to Saudi Arabia in the methods. I still think some information in the introduction is therefore required detailing the costs of COVID-19 specifically to the Saudi population and how healthcare utilisation was perhaps effected during the pandemic. It is acknowleged that COVID is a global pandemic, but some countries have had fewer cases and deaths by using appropriate control strategies. From reading your article it is not known what impact COVID-19 had/is having specifically in SA. I would suggest adding a sentence or two with some references. I don't get a sense of the COVID cases, hospitalisations or deaths from refs 1-3 included in the document at the moment for example. This information will enable the reader to better determine the SA need for the technology discussed.

Author Response

  • Response to reviewer 1

    • Thank you for your valuable comments. Your inputs have helped strengthen both the presentation and the quality of the paper. We revised the background and introduced COVID-19 disruption to the routine health care services. It reads now “Responses to the pandemic have varied dramatically, and the demand on healthcare professionals has increased due to the disruption caused by COVID-19 outbreak” Lines 33-35.

Reviewer 2 Report

This article aims to sumarize an experience with digital health platforms in the follow-up and control of COVID-19 patients. It provides novel information about the effectiveness of this methods and specially in the situation of pandemic we are living this information could be useful for other health care systems. Furthermore, the paper provides concise information. Some suggestions are detailed below:

- In the methods section, if the type of subjects included has been considered, please detail it in the inclusion/exclusion criteria.

- All of the information provided suggest that this DHPTs have been used to control COVID-19 outbreaks but some information at the end of the results or in the conclusion section about the usefulness of this methods to control long-covid patients or the development of the acute cases could be interesting.

In summary, the manuscript is very interesting and very well written, and only some considerations have been made since, in the opinion of this reviewer, no changes are needed to be published.

Author Response

Response for reviewer 2

  • Thank you for the feedback on the manuscript. Very much appreciated. Your inputs have helped strengthen both the presentation and the quality of writing. We provided sufficient description for the inclusion and exclusion articles in online supplementary. We appreciated the suggestion about long-COVID management. This is a really good idea of research, and it could be done in the future by evaluating long-term acceptability of DHTPs.